# Reinforcing Agentic Search Via Reward Density Optimization

## Abstract

Reinforcement Learning with Verifiable Rewards (RLVR) is a promising approach for enhancing agentic deep search. However, its application is often hindered by low **Reward Density** in deep search scenarios, where agents expend significant exploratory costs for infrequent and often null final rewards. In this paper, we formalize this challenge as the **Reward Density Optimization** problem, which aims to improve the reward obtained per unit of exploration cost. This paper introduce **InfoFlow**, a systematic framework that tackles this problem from three aspects. 1) **Sub-goal Scaffolding**: breaking down long-range tasks to assign process rewards, thereby providing denser learning signals. 2) **Pathfinding Hints**: injecting corrective guidance into stalled trajectories to increase the probability of successful outcomes. 3) **Dual-agent refinement**: employing a dual-agent architecture to offload the cognitive burden of deep exploration. A refiner agent synthesizes the search history, which effectively compresses the researcher's perceived trajectory, thereby reducing exploration cost and increasing the overall reward density. We evaluate InfoFlow on multiple agentic search benchmarks, where it significantly outperforms strong baselines, enabling lightweight LLMs to achieve performance comparable to advanced proprietary LLMs. Our codes are in *this repository*.

## 1 Introduction

Large language models (LLMs) have become essential tools for information seeking in daily life (Zhao et al., 2023; Gao et al., 2023). As their applications expand, users increasingly expect LLMs to handle not only factual queries but also complex, multi-step tasks requiring knowledge discovery and synthesis. However, because an LLM's internal knowledge is limited and quickly outdated, relying solely on parametric memory is insufficient for knowledge-intensive tasks (Vu et al., 2023). Addressing such challenges requires integrating external knowledge sources and moving beyond surface-level retrieval toward deeper reasoning and information synthesis (Shi et al., 2023). Most existing approaches follow the retrieval-augmented generation (RAG) paradigm (Gao et al., 2023), which treats the input as a query and retrieves relevant documents for generation. While effective for factual questions, RAG struggles with hierarchical or implicit information needs (Asai et al., 2023; Qian et al., 2025). Extensions such as query rewriting, iterative retrieval, and self-refinement (Ma et al., 2023; Jiang et al., 2023; Madaan et al., 2023) improve flexibility but remain bound to a *pre-inference* design that retrieves information before reasoning begins, limiting adaptability in dynamic, multi-step tasks.

Inspired by reasoning-centric models (OpenAI, 2024; DeepSeek-AI, 2025), recent studies adopt the *search-integrated reasoning* (SIR) paradigm (Yao et al., 2023; Chen et al., 2025a; Xue et al., 2025; Huang et al., 2025), which interleaves reasoning and search to adaptively incorporate external knowledge at each step (Li et al., 2025c; Jin et al., 2025b; Li et al., 2025b). However, current LLMs lack native mechanisms to invoke external search tools. Early SIR implementations relied on manually crafted prompts and exhibited limited generalization (Li et al., 2025a). To overcome this, *Reinforcement Learning with Verifiable Rewards* (RLVR) has emerged as an effective approach for training LLMs to conduct agentic deep search. RLVR enables models to learn search-integrated reasoning policies via trajectory rollouts and final reward-driven optimization (Jin et al., 2025b;a; Qian & Liu, 2025).

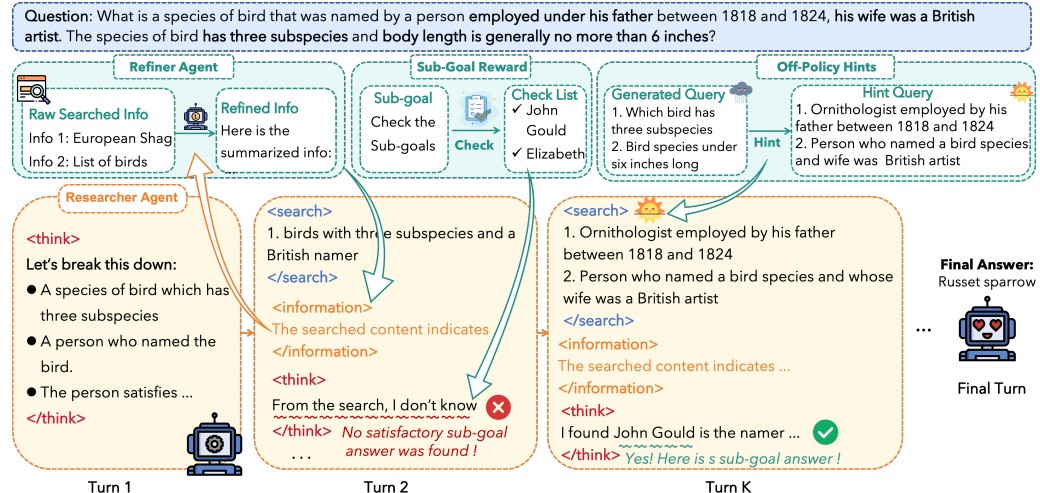

Figure 1: The framework of InfoFlow and example of DSQA task. Researcher agent focuses on reasoning and planning, refiner agent synthesizes massive searched content into condensed info.

Despite its promise, RLVR for deep search suffers from **low reward density**, which we define as total reward per unit exploration cost (i.e., per trajectory length). Deep search tasks typically require multiple turns of reasoning-searching exploration before producing a final answer. As *trajectory length increases*, success rates decline rapidly, since a single reasoning error can accumulate to invalidate the entire trajectory. Moreover, recent work highlights that training robust search agents *requires more complex, reasoning-intensive tasks* (Xia et al., 2025; Tao et al., 2025; Bae et al., 2025; Yan et al., 2025). However, our preliminary experiments (Fig. 2) show that on difficult tasks successful rollouts become rare (often less than 10% of initial accuracy), further reducing reward density and increasing computational inefficiency.

To address these issues, we formulate **Reward Density Optimization** and propose **InfoFlow**, a reinforcement learning framework that improves reward accessibility and stabilizes learning in search-integrated reasoning. InfoFlow increases reward density and learning efficiency via three core components: **(1) Sub-goal Scaffolding.** To make deep search more tractable for agents with limited initial capabilities, InfoFlow decomposes complex search queries into sub-goals and awards intermediate rewards for solving them. Deep search tasks naturally exhibit hierarchical structure: reaching the final answer typically requires identifying intermediate key facts or anchor entities. Rather than assigning rewards only for full task success, InfoFlow grants partial rewards for resolved sub-goals, providing denser feedback for policy updates. This scaffolding yields a denser learning signal and mitigates the sparsity of final rewards. **(2) Pathfinding Hints.** To guide agents toward full solutions, InfoFlow incorporates explicit guidance during RL exploration in the form of pathfinding hints. We employ LLM (Gemini 2.5 or Qwen3-8B (Gemini Team, 2025; Yang et al., 2025)) as annotators to enrich training data (§ A.2) by generating search queries that guide the agent toward reaching key sub-goals. When the agent struggles to reach final answers within a predefined turns during on-policy rollouts, InfoFlow inserts guiding queries into the next turn to suggest more informative search directions. These pathfinding hints make intermediate key facts and anchor entities easier to discover, increasing sub-goal rewards and the likelihood of a correct final answer. They also help the agent learn improved search strategies via learning from expert demonstrations. **(3) Dual-agent refinement.** To reduce the cognitive burden associated with long trajectories, InfoFlow adopts a dual-agent design for deep search. A *research agent* performs reasoning and search, while a *refiner agent* condenses retrieved information into concise, structured summaries that are fed back to the research agent. This collaboration improves efficiency and accuracy: we observe up to 59.5% higher initial rewards, 16.4% reduced inference time, and 44.8% shorter trajectories (§ 3.1), substantially increasing reward density. Together, these techniques enable InfoFlow to overcome the reward sparsity bottleneck in RL training, making complex information-seeking tasks more tractable for agents while fostering deeper reasoning and evidence synthesis.

We evaluate InfoFlow on a suite of knowledge-intensive agentic search benchmarks as well as the challenging complex benchmark BrowseComp-Plus (Chen et al., 2025b). Experimental results demonstrate that our method consistently outperforms strong baselines. Notably, on BrowseComp-

Plus, our optimized small-scale model achieves performance comparable to much larger LLMs. Our main contributions are summarized as follows: (1) We propose InfoFlow, a dual-agent framework for agentic deep search, where a researcher agent is responsible for central reasoning and planning, while a refiner agent synthesizes retrieved evidence into coherent knowledge. (2) We introduce a well-tailored reward density optimization strategy, comprising sub-goal reward shaping, adaptive off-policy hints, and supervised initialization via reject sampling. These techniques collectively alleviate reward sparsity and stabilize training. (3) Through extensive experiments, we verify the effectiveness of InfoFlow. In particular, on the challenging BrowseComp-Plus benchmark, InfoFlow enables a small-scale model to achieve performance competitive with much larger LLMs.

## 2 PRELIMINARY AND DATA PREPARATION

### 2.1 DEEP SEARCH QUESTION ANSWERING

The task of **Deep Search Question Answering (DSQA)** involves addressing complex queries that require multi-step reasoning and extensive information seeking. Benchmarks such as *BrowseComp* (Wei et al., 2025) exemplify this challenge, evaluating agentic search capability to navigate large-scale corpora such as the internet and synthesize information into coherent answers.

To enable principled optimization, we formalize DSQA as a reasoning tree $\mathcal{T}$, following the framework of Xia et al. (2025). In this formulation, each node denotes a sub-problem, either an entity to be identified or a constraint imposed on its parent entity. The root node is the final answer to the DSQA problem, which is a fact or entity to be discovered. Directed edges from child to parent nodes encode logical dependencies that must be validated. The complexity of DSQA problem is characterized by two structural properties of the reasoning tree. The *depth*, defined as the length of the longest root-to-leaf path, captures the extent of sequential reasoning required to resolve all sub-problems. The overall *width*, measured as the sum of children across all non-leaf nodes, reflects the degree of parallel information aggregation necessary to complete the task.

### 2.2 FORMULATION OF AGENTIC DEEP SEARCH PROCESS

The process of an LLM agent solving DSQA task can be formalized as a Markov Decision Process (MDP) (Puterman, 1990). An agent's trajectory $\tau$ is a sequence of interactions with a search environment: $\tau = (q, a_0, i_0, a_1, i_1, \ldots, a_{K-1}, i_{K-1}, a_K)$. Here, $q$ is the initial question, $a_k$ is the agent's action at step $k$, $i_k$ is the information retrieved from the environment, and $a_K$ is the terminal action containing the final answer. The MDP is defined by the tuple $(\mathcal{S}, \mathcal{A}, \mathcal{P}, \mathcal{R}, \gamma)$, where:

**Action** $(a_k)$. The agent generates an action $a_k$, which involves two components: *Thinking* $(a_k^{think})$: A reasoning trace where the agent analyzes the current state $S_k$, synthesizes retrieved knowledge, and plans its next steps. This corresponds to *depth-wise progress* in the reasoning tree by exploiting available information and is enclosed in `<think>` tags. *Searching* $(a_k^{search})$: The agent generates a set of $N_k$ parallel search queries $\{q_{k,j}\}_{j=1}^{N_k}$ to acquire new information. This facilitates *width-wise exploration* of the reasoning tree and is enclosed in `<search>` tags. The full action is the concatenation $a_k = a_k^{think} \circ a_k^{search}$. The action space also includes a terminal action $a_K$, where the agent provides the final answer within `<answer>` tags.

**Transition** $(\mathcal{T})$. The transition function $\mathcal{P}(S_{k+1}|S_k, a_k)$ is determined by the environment's response to the search action. An external search tool processes the queries $\{q_{k,j}\}$ and returns a set of retrieved evidence $i_k = \{(q_{k,j}, e_{k,j})\}_{j=1}^{N_k}$. This information is presented to the agent within `<information>` tags. The subsequent state is formed by appending the action and observation to the history: $S_{k+1} = S_k \circ (a_k, i_k)$.

**Reward** $(R)$. A final reward $R(\tau)$ is assigned based on the correctness of the final answer $a_K$, evaluated by a rule-based reward model. The agent's objective is to learn a policy $\pi(a|S)$ that maximizes the expected return.

### 2.3 DATA PREPARATION WITH ENRICHED PROCESS INFORMATION

As described in introduction, optimizing agentic search via RL is challenged by low reward density. This problem is particularly pronounced in complex deep search tasks, where agents must execute long exploratory trajectories. Since agentic RL methods depend on outcome-based rewards (Dong et al., 2025; Jin et al., 2025b; Sun et al., 2025), the rarity of success often leaves agents with no

feedback after costly exploration, making policy gradient methods ineffective on predominantly unsuccessful trajectories.

We argue that this sparsity arises from a lack of training data with dense, process-level supervision. To address this gap, we build on the open-source *InfoSeek* dataset (Xia et al., 2025). Unlike datasets such as Natural Questions or HotpotQA (Kwiatkowski et al., 2019a; Yang et al., 2018), which emphasize single- or two-hop reasoning, InfoSeek is designed for multi-step information seeking, providing a more suitable foundation for our work.

We enrich the 18,000 training instances in *InfoSeek* (Xia et al., 2025) with two forms of off-policy supervision, generated using the Gemini 2.5 API (Gemini Team, 2025). This augmented data is designed to directly facilitate the reinforcement learning strategies detailed in § 3: (1) **Sub-goal Scaffolding:** For each problem's reasoning tree, we use LLM (Gemini 2.5 and Qwen3-8B) to annotate and select only the most informative nodes representing critical entities as distinct sub-goals. These entities constitute mandatory milestones, as their identification represents significant breakthroughs essential for resolving the overall query. We form a ground-truth set of sub-goals $\mathcal{G}_q = \{g_1, \ldots, g_M\}$ and annotate each sub-goal $g_i$ with a normalized importance weight $s_i$, reflecting their contribution for solving the overall deepsearch task. The weights are constrained to sum to one: $\sum_{i=1}^{M} s_i = 1$. The final enriched data thus provides a set of weighted sub-goals $\{(g_i, s_i)\}_{i=1}^{M}$ for each question, enabling a granular basis for the sub-goal reward shaping scheme used to encourage structured decomposition during RL. (2) **Pathfinding Hints:** To lower the exploration barrier for particularly difficult reasoning steps, we generate hints for **critical edges** in the reasoning tree. We employ LLMs for hint annotation; empirically, both proprietary (e.g., Gemini 2.5) and open-source models (e.g., Qwen3-8B) are capable of effectively performing this annotation task. Unlike simple keyword prompts, these hints are formulated as high-leverage **guiding queries** that decompose intertwined constraints into actionable search steps. They are designed to teach the agent three specific skills: *purposeful search* for specific sub-problems, *bottleneck breakthrough* for non-obvious reasoning points, and *creative search* via constraint reframing. These pre-generated queries act as information bridges, providing adaptive off-policy guidance to mitigate unproductive exploration loops during on-policy RL (Yan et al., 2025; Zhang et al., 2025; Wu et al., 2025). Further details on the prompt construction and concrete examples are provided in Appendix § A.2.

## 3 METHOD

We formally define reward density as the expected reward obtained per unit exploration cost, which reflects how efficiently a search agent transforms exploratory computation into verifiable learning signals. Given a dataset of $n$ deep search QA instances, each solved by a leading search agent coupled with an external search engine, we conduct $k$ rollouts per instance under a non-zero sampling temperature to ensure exploration diversity. For the $j$-th rollout of instance $i$, we denote the final reward as $r_{i,j} \in \{0, 1\}$, indicating correctness of the final answer, and the trajectory length as $l_{i,j}$, representing the trajectory length of the search agent. The reward density $\tau$ is computed as:

$$\tau = \frac{\sum_{i=1}^{n} \sum_{j=1}^{k} r_{i,j}}{\sum_{i=1}^{n} \sum_{j=1}^{k} l_{i,j}}.$$

Reward density is the key to the efficiency and scalability of both Rejection sampling Fine-Tuning (RFT) and Reinforcement Learning (RL) stages, which constitute the common two-phase optimization paradigm for search agents. Higher $\tau$ provides more successful trajectories for supervised learning in RFT and stronger, more stable gradient signals for policy optimization in RL.

**InfoFlow** addresses the challenge of *low reward density* in deep search training by formulating learning as a **Reward Density Optimization** problem. We enhance the reward density through three comprehensive and complementary mechanisms: (i) **Sub-goal Scaffolding** (dense, process-level rewards; see § 3.3), (ii) **Pathfinding Hints** (adaptive off-policy guidance; see § 3.4), and (iii) **Dual-agent refinement** (dual-agent compression of retrieved evidence; see § 3.1).

### 3.1 DUAL-AGENT REFINEMENT

The cognitive burden of managing long, noisy trajectories in deep search is a key driver of low reward density. To mitigate this, our framework (Figure 1) decouples this process into a ***Researcher Agent*** ($\pi_\theta$) for planning and exploration, and a ***Refiner Agent*** ($\mathcal{F}_\phi$) for information synthesis.

The *Researcher* navigates the reasoning tree by generating actions $a_k = a_k^{\text{think}} \circ a_k^{\text{search}}$, where $a_k^{\text{search}}$ can issue parallel queries $\{q_{k,j}\}_{j=1}^{N_k}$ to explore multiple lines of inquiry. For each query, the *Refiner* (driven by a LLM described in § A.6.1) processes the resulting noisy evidence $e_{k,j}$ and distills it into a concise summary: $sum_{k,j} = \mathcal{F}_\phi(q, q_{k,j}, e_{k,j})$. These summaries form the structured information $i_k = \{(q_{k,j}, sum_{k,j})\}_{j=1}^{N_k}$ that updates the researcher's state to $S_{k+1}$.

The advantage of the decoupled architecture lies in its ability to enhance the **reward density** (higher accuracy with less context length), which lays the foundation for later stable on-policy RL. As shown in Figure 2, we conduct experiments on InfoSeek evaluation set using Qwen2.5-3B-Instruct as the researcher, and compare varying refiner configurations. The introduction of a 3B refiner improves the success rate by 5.0 points while reducing the researcher's context length by nearly 45% (from 2372 to 1310 tokens). The context reduction frees up the researcher's limited context window to focus on high-level reasoning and planning rather than being overwhelmed by verbose, unprocessed evidence. More detailed efficiency analysis is conducted in § A.3.

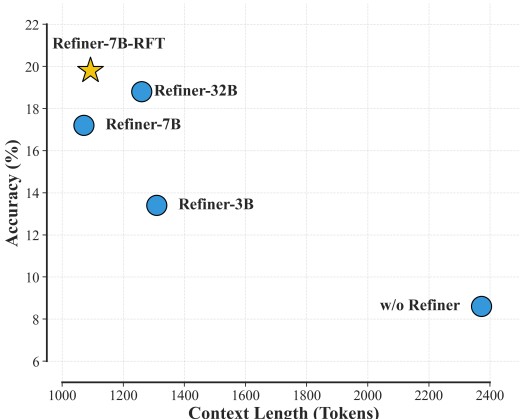

Figure 2: Dual agent framework enhances reward density: achieving higher accuracy with less context.

### 3.2 REJECTION SAMPLING FINE-TUNING FOR REWARD-DENSE INITIALIZATION

Preliminary experiments in Figure 2 show less than 10% accuracy for untrained agents, yielding extremely sparse rewards. To mitigate this cold-start issue, we construct a high-quality corpus using rejection sampling and use it to jointly fine-tune both the Researcher and Refiner.

**Trajectory collection and verification.** We start from 18,000 DSQA tasks in the *InfoSeek* dataset (Xia et al., 2025). Using the base dual-agent framework (Qwen2.5-7B-Instruct for both roles), we perform two rollouts per task and retain only trajectories that produce correct final answers. We then apply a powerful verifier (Gemini-2.5-Pro (Gemini Team, 2025)) to filter out trajectories that succeed by chance or contain flawed reasoning; the final corpus contains ≈3,450 high-quality trajectories. This corpus encodes step-level reasoning and search-grounded evidence, providing dense supervised signals absent in standard pretraining data.

**Joint fine-tuning objective.** We co-train the Researcher policy $\pi_\theta$ and the Refiner $\mathcal{F}_\phi$ on the verified trajectories. The Researcher is trained with token-level negative log-likelihood on demonstrated actions:

$$\mathcal{L}_{\text{SFT}}^{\text{researcher}}(\theta; \tau) = -\sum_{k=0}^{K} \sum_{t=1}^{|a_k|} \log \pi_\theta(a_{k,t} \mid S_k, a_{k,<t}),$$

while the Refiner is trained to map raw evidence to compact summaries:

$$\mathcal{L}_{\text{SFT}}^{\text{refiner}}(\phi; \tau) = -\sum_{k=0}^{K-1} \sum_{j=1}^{N_k} \log P_\phi(sum_{k,j} \mid q, q_{k,j}, e_{k,j}).$$

Joint RFT yields a substantially higher initial success rate and reduces trajectory verbosity, making subsequent RL more stable. Empirical comparisons are reported in Table 1.

### 3.3 SUB-GOAL SCAFFOLDING

After the RFT initialization, we conduct RLVR to further enhance deep search capability of InfoFlow. The sparse-reward challenge in deep search RL arises from both task complexity and outcome-based reward. Single binary reward for final answer offers limited guidance for the intermediate steps of a long reasoning trajectory, particularly when early-stage success rates are low.

Table 1: Analytical experiments on InfoSeek eval set. Co-training (RFT on both) improves Mean@4 and reduces the fraction of unsolved ("Solve None") samples.

| Configuration | Mean@4 | Solve None(%) | Context (Tok.) | Search Calls |
|---|---|---|---|---|
| Base Agents (researcher-3B + refiner-7B) | 17.2 | 76.7 | 1071.2 | 2.83 |
| + RFT on researcher only | 31.0 | 50.3 | 2489.3 | 3.92 |
| + RFT on Both (Co-training) | 34.3 | 46.0 | 2612.0 | 4.17 |

To provide informative learning signals inside long trajectories, we decompose each complex question into a set of weighted sub-goals $\{(g_i, s_i)\}_{i=1}^{M}$ (e.g., find anchor entities, verify key facts) as described in § 2.3. For a trajectory $\tau$, let $\mathcal{G}_{\text{solved}}(\tau)$ denote the sub-goals resolved by the agent. We define a process-level reward

$$R_{\text{sub}}(\tau) = \sum_{g_i \in \mathcal{G}_{\text{solved}}(\tau)} s_i,$$

with $\sum_i s_i = 1$. The total trajectory reward combines the binary final reward and the sub-goal reward:

$$R(\tau) = R_{\text{final}}(\tau) + w \cdot R_{\text{sub}}(\tau),$$

where $w$ trades off final correctness and intermediate progress (we use $w = 0.3$). This shaped reward provides gradient information for partially correct trajectories and encourages decomposed reasoning.

### 3.4 PATHFINDING HINTS

While sub-goal rewards densify the learning signal, on-policy exploration alone remains a bottleneck for the more challenging problems. Even after RFT, a significant portion of difficult samples are never solved through multiple rollouts (as suggested by the solve none ratio with four rollouts in pilot analytical studies Table 1), hindering learning signal to policy gradient updates. This is because the agent can become trapped in unproductive exploration loops, failing to discover the critical reasoning paths necessary for success.

To overcome this exploration barrier, we introduce *pathfinding hints* to provide help during on-policy rollouts. We leverage the guiding queries prepared in § 2.3, which are high-leverage search actions designed to bridge difficult logical steps. The pathfinding hints injection is triggered when a trajectory exceeds a predefined turn threshold, $K_h$, without reaching a terminal state. At this step $(k = K_h)$, the executed action $a'_k$ is constructed by combining the agent's original reasoning trace $a_k^{\text{think}}$ with the pre-constructed hint queries $a_{k,\text{hint}}^{\text{search}}$:

$$a'_k = a_k^{\text{think}} \circ a_{k,\text{hint}}^{\text{search}}. \tag{1}$$

The agent then receives the information retrieved using these hint queries and continues its trajectory from the new state. We set $K_h = 5$ in practice.

This mechanism offers two-fold benefits for stabilizing RL. First, as an exploration corrective, it rescues the agent from unproductive loops, increasing the yield of successful trajectories essential for policy optimization. Second, as an explicit demonstration, it exposes the agent to an informative off-policy, expert-quality search action and its positive outcome for better learning.

### 3.5 POLICY OPTIMIZATION

We fine-tune the researcher via reinforcement learning that integrates the shaped reward $R(\tau)$ and hint-guided exploration after RFT. We adopt Group Relative Policy Optimization (GRPO) (Shao et al., 2024), a PPO-style algorithm that normalizes advantages within trajectory groups to reduce variance. For a batch of $G$ trajectories $\{\mathcal{Y}_i\}$ with returns $\{R_i\}$, the group-normalized advantage is

$$A_i = \frac{R_i - \text{mean}(\mathbf{R})}{\text{std}(\mathbf{R})},$$

and the GRPO objective is

$$\mathcal{J}_{\text{GRPO}}(\theta) = \mathbb{E}\left[\frac{1}{G}\sum_{i=1}^{G}\min\Big(r_i A_i,\ \text{clip}(r_i, 1-\epsilon, 1+\epsilon)A_i\Big) - \beta\, \mathrm{D}_{\text{KL}}(\pi_\theta \,\|\, \pi_{\text{ref}})\right],$$

where $r_i$ is the importance ratio and $\pi_{\text{ref}}$ is a reference policy used for KL regularization.

## 4 EXPERIMENTS

In this section, we empirically validate InfoFlow. Our experiments are designed to demonstrate that by systematically optimizing for *reward density*, our framework achieves strong performance and generalization for agentic search tasks, particularly on complex deep search tasks.

### 4.1 EXPERIMENTAL SETUP

**Datasets and Evaluation Metrics.** To assess the **general information-seeking and agentic search** capability, we test InfoFlow on a suite of widely-used single-hop and multi-hop QA benchmarks with external search corpus: Natural Questions (NQ) (Kwiatkowski et al., 2019b), TriviaQA (TQA) (Joshi et al., 2017), PopQA (Mallen et al., 2022), HotpotQA (HQA) (Yang et al., 2018), 2WikiMultihopQA (2Wiki) (Ho et al., 2020), Musique (MSQ) (Trivedi et al., 2022), and Bamboogle (Bamb) (Press et al., 2022). We use E5 (Wang et al., 2024) as the embedding model, the 2018 Wikipedia dump (Karpukhin et al., 2020) as the corpus, and set the number of retrieved passages to 3. We report Exact Match (EM) as the metric for these datasets. To evaluate **deep search capability**, we employ the BrowseComp-Plus benchmark (Chen et al., 2025b), a refined version of BrowseComp (Wei et al., 2025) with 830 challenging problems and a fixed 100K webpage corpus. This benchmark is an ideal testbed for DSQA as its problems inherently demand the deep, iterative reasoning and search. Following the official implementation, accuracy is judged by an LLM (we use deepseek v3.1 (DeepSeek-AI, 2024) to judge).

**Baselines and Implementation Details.** We compare InfoFlow against recent agentic search methods, including Self-RAG (Asai et al., 2023), Search-o1 (Li et al., 2025c), Search-R1 (Jin et al., 2025b), Zero-Search (Sun et al., 2025), AutoRefine (Shi et al., 2025), InForage (Qian & Liu, 2025), and ParrallelSearch (Zhao et al., 2025). These methods employ multi-turn interactions but differ in their training strategies and agentic framework. For the complex BrowseComp-Plus benchmark, we include proprietary models like Gemini 2.5 Pro (Comanici et al., 2025), Sonnet 4 (Anthropic, 2025), GPT-5 (OpenAI, 2025), and larger open-sourced Qwen3-32B (Yang et al., 2025) and Search-R1-32B (Jin et al., 2025b). Our model is initialized with the framework described in § 3.1. For InfoFlow-3B and InfoFlow-7B, we use Qwen2.5-3B-Instruct/Qwen2.5-7B-Instruct (Group, 2025) as initialization for researcher agent respectively. We use Qwen2.5-7B-Instruct as initialization for refiner agent. Then InfoFlow is trained using the pipeline detailed in § 3. Further training details are provided in § A.4.

### 4.2 MAIN RESULTS

#### 4.2.1 INFOFLOW DEMONSTRATES SUPERIOR GENERALIZATION ON QA TASKS

As shown in Table 2, InfoFlow demonstrates strong performance and generalization ability on standard agentic search and information-seeking benchmarks, outperforming all baseline models at both the 3B and 7B scales. Unlike baseline methods, which primarily rely on in-domain training data such as NQ and HQA, InfoFlow maintains robust and transferable performance without requiring in-domain supervision. This result highlights the effectiveness of our reward density optimization approach with the enriched InfoSeek dataset, which encourages more resilient and generalizable reasoning by providing dense, process-level rewards. These rewards enable the model to capture the compositional structure of multi-step reasoning. The benefit is particularly evident on multi-hop datasets such as HQA and 2Wiki, where the method explicitly trains the agent to synthesize information step by step, a critical capability for complex information-seeking tasks. These rewards enable the model to capture the compositional structure of multi-step reasoning. The benefit is particularly evident on multi-hop datasets such as HQA and 2Wiki, where the method explicitly trains the agent to synthesize information step by step, a critical capability for complex information-seeking tasks.

Table 2: Performance comparison on QA tasks with agentic search methods. The best result in each column is highlighted in **bold**.

| Model | NQ | TQA | PopQA | HQA | 2Wiki | MSQ | Bamb | Avg. |
|---|---|---|---|---|---|---|---|---|
| *Qwen2.5-3B Based Models* | | | | | | | | |
| Search-o1-3B | 23.8 | 48.2 | 26.2 | 22.1 | 21.8 | 5.4 | 32.0 | 25.6 |
| Search-R1-3B | 40.8 | 59.1 | 42.8 | 30.8 | 31.1 | 8.4 | 13.0 | 32.3 |
| ZeroSearch-3B | 41.2 | 61.5 | 44.0 | 31.2 | 33.2 | 12.6 | 14.3 | 34.0 |
| AutoRefine-3B | 43.6 | 59.7 | 44.7 | 40.4 | 38.0 | 16.9 | 33.6 | 39.6 |
| InForage-3B | 42.1 | 59.7 | 45.2 | 40.9 | 42.8 | 17.2 | 36.0 | 40.6 |
| ***InfoFlow-3B*** | **44.5** | **63.7** | **47.0** | **44.6** | **45.2** | **21.0** | **41.2** | **43.9** |
| *Qwen2.5-7B Based Models* | | | | | | | | |
| Self-RAG-7B | 36.4 | 38.2 | 23.2 | 15.7 | 11.3 | 3.9 | 5.6 | 19.2 |
| Search-o1-7B | 27.7 | 47.4 | 29.4 | 34.8 | 35.6 | 4.8 | 15.2 | 27.1 |
| Searcn-R1-7B | 38.3 | 59.3 | 39.9 | 37.6 | 31.7 | 15.1 | 38.1 | 37.0 |
| ZeroSearch-7B | 43.6 | 65.2 | **48.8** | 34.6 | 35.2 | 18.4 | 27.8 | 39.1 |
| ParallelSearch-7B | 46.2 | 62.8 | 42.9 | 42.9 | 42.4 | 19.7 | 41.1 | 42.5 |
| ***InfoFlow-7B*** | **47.2** | **68.1** | 48.1 | **44.3** | **47.2** | **21.9** | **47.6** | **46.2** |

### 4.2.2 INFOFLOW EXCELS AT COMPLEX LONG-HORIZON DEEP SEARCH TASKS

We conduct evaluation on BrowseComp-Plus to test the deep information seeking capability of InfoFlow. For fair comparison, all models use BM25 as retriever. As shown in Table 3, InfoFlow substantially outperforms existing open-source agents, even those based on larger 32B models. Notably, it also surpasses strong proprietary models like Gemini 2.5 Pro and GPT-4.1. The dual-agent framework preserves the researcher's focus on high-level strategic planning. Concurrently, our data-centric RL approach (§ 3.4), which uses sub-goal rewards and adaptive hints, provides the dense and structured supervision necessary to navigate complex reasoning paths where sparse rewards would otherwise stall learning, thus making InfoFlow effectively

Table 3: Performance and search calls on the complex BrowseComp-Plus benchmark.

| Model | Accuracy (%) | Search Calls |
|---|---|---|
| Gemini 2.5 Flash | 15.5 | 10.6 |
| Gemini 2.5 Pro | 19.0 | 7.4 |
| Sonnet 4 | 14.3 | 10.0 |
| GPT-4.1 | 14.6 | 11.2 |
| GPT-5 | 55.9 | 23.2 |
| Qwen3-32B | 3.5 | 0.9 |
| SearchR1-32B | 3.9 | 1.8 |
| *InfoFlow-3B* | 18.5 | 8.1 |
| *InfoFlow-7B* | 23.2 | 7.9 |

solving difficult deep search tasks.

Table 4: Ablation study of InfoFlow components. We report average accuracy on seven general QA tasks, accuracy on the BrowseComp-Plus. and InfoSeek-Eval benchmarks.

| Configuration | QA Average | BrowseComp-Plus | InfoSeek-Eval |
|---|---|---|---|
| InfoFlow-7B | 46.2 | 23.2 | 47.8 |
| *w/o* Dual-Agent RFT | 38.4 | 10.2 | 32.5 |
| *w/o* Sub-Goal Reward | 44.9 | 21.4 | 44.5 |
| *w/o* Off-Policy Hints | 45.8 | 20.1 | 42.1 |

### 4.3 DISCUSSION

#### 4.3.1 ABLATION STUDY

We perform ablations on InfoFlow-7B to evaluate the contribution of each component: (1) Removing **dual-agent RFT** causes the largest performance degradation. The combination of low success

rates and long trajectories results in extremely low reward density, which are insufficient for stable policy optimization. (2) Removing **sub-goal reward shaping** also yields a consistent decrease. This finding underscores the importance of dense intermediate supervision for on-policy RL. (3) Without **off-policy hints** has a relatively minor effect on general QA but leads to a 3.1-point drop on BrowseComp-Plus, indicating that hints are especially valuable for difficult information-seeking tasks requiring deep search, intensive reasoning, and long-horizon exploration.

### 4.3.2 ANALYSIS OF REASONING DEPTH

We conduct experiments to analyze how InfoFlow's performance scales with reasoning depth on the challenging BrowseComp-Plus benchmark. As shown in Figure 3, allowing more reasoning-searching turns improves accuracy effectively, which increases from 11.2% (4 turns) to 22.8% (16 turns). This result demonstrates that InfoFlow learns a generalizable, iterative reasoning policy rather than being limited by the fixed max reasoning-searching turns during training. This allows the agent to dynamically extend its reasoning process during inference, a crucial capability for deep search tasks where the required reasoning depth to be adaptively adjusted.

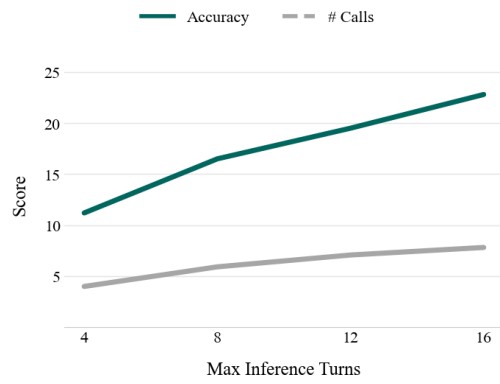

Figure 3: Analysis of Reasoning Depth.

### 4.3.3 REINFORCEMENT LEARNING TRAINING DYNAMICS

We examine the RL training dynamics of InfoFlow-7B with and without sub-goal shaping and hints. We report both the original final reward (task accuracy, green curve) and the shaped reward (pink curve). The two curves improve in tandem rather than diverging. If reward hacking were present, the shaped reward would increase while the final reward stagnated or declined. Instead, both metrics rise consistently, indicating that the agent is learning genuinely improved search behaviors rather than exploiting annotation.

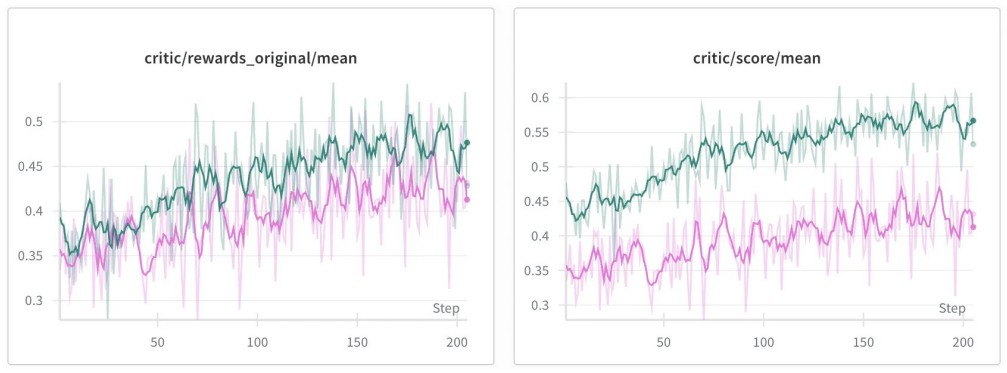

Figure 4: RL training dynamics with and without hints and sub-goal rewards.

## 5 CONCLUSION

We introduced InfoFlow, a dual-agent framework designed to address the critical challenge of low reward density in training LLM agents for agentic deep search tasks. By integrating sub-goal reward shaping, adaptive off-policy hints, and a dual-agent architecture initialized with RFT, InfoFlow provides dense, process-level supervision that makes learning tractable. Our experiments demonstrate that this approach enables even lightweight LLMs to achieve performance competitive with much larger proprietary models on challenging deep search benchmarks. This work highlights the efficacy of data-centric RL in making complex agentic deep search tractable and presents a promising direction for developing more capable and efficient LLM search agents.

## REPRODUCIBILITY STATEMENT

To ensure the reproducibility of our research, we provide a detailed account of our methodology and experimental setup. Our code, along with the enriched InfoSeek dataset, will be made publicly available upon publication. The experimental setup, including datasets, evaluation metrics, and baseline models, is described in § 4.1. Key implementation details and hyperparameters for our proposed InfoFlow framework are presented throughout § 3. Specifically, the dual-agent RFT process is detailed in § 3.2, and the reinforcement learning approach, including the sub-goal reward weight ($w = 0.3$) and the hint injection threshold ($K_h = 5$). Detailed hyperparameters and further implementation details are provided in the Appendix.

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

# A APPENDIX

## A.1 RELATED WORK

**From Retrieval Augmentation to Search-Integrated Reasoning.** To mitigate the limitations of static parametric knowledge, Retrieval-Augmented Generation (RAG) has become a standard practice (Lewis et al., 2020). Early RAG methods follow a static "retrieve-then-generate" pipeline, which struggles with complex, multi-hop queries. Recent efforts have made this process more dynamic through query rewriting, iterative retrieval, or self-critique mechanisms that assess the relevance of retrieved information (Asai et al., 2024). A more advanced paradigm, Search-Integrated Reasoning (SIR), moves beyond this separation by deeply interleaving reasoning steps with tool actions like web searches. Foundational frameworks such as ReAct (Yao et al., 2023) demonstrated the effectiveness of this approach using in-context learning. Our work, InfoFlow, adopts the SIR paradigm but focuses on explicitly training models to acquire these capabilities, rather than relying solely on prompt engineering at inference time.

**Training Agents for Search and Reasoning.** A prominent research direction focuses on fine-tuning LLMs to learn robust policies for interacting with search engines. While Supervised Fine-Tuning (SFT) on expert trajectories provides a strong initialization (Zeng et al., 2023), Reinforcement Learning (RL) is crucial for teaching agents to explore and discover effective strategies for unseen problems. Several works have successfully applied RL to train search agents (Jin et al., 2025b; Song et al., 2025). However, a fundamental obstacle is reward sparsity: complex tasks yield infrequent terminal rewards, providing poor learning signals for the long sequence of intermediate steps (Ning et al., 2025). This makes policy optimization unstable and inefficient. While some methods attempt to mitigate this by learning a separate reward model or using offline policy optimization (Wang et al., 2025; Deng et al., 2025), InfoFlow addresses the problem directly through a novel combination of sub-goal reward shaping to provide dense, intermediate signals and adaptive off-policy hints to increase the rate of successful trajectory completion during online training.

**Multi-Agent Collaboration.** Decomposing complex problems for multi-agent systems is a powerful strategy. Most current approaches focus on inference-time orchestration, where a central planner LLM delegates sub-tasks to specialized tools or other LLM instances without altering their weights (Qiu et al., 2025). Frameworks like MetaGPT (Hong et al., 2023) assign distinct roles to different LLM agents to collaboratively solve complex tasks. InfoFlow advances this concept by introducing a co-trained dual-agent framework. We partition the cognitive load between a Researcher agent for planning and execution and a Refiner agent for evidence synthesis and guidance. Crucially, unlike inference-time frameworks, our agents are jointly optimized, allowing them to develop a specialized and synergistic protocol that enhances reasoning efficiency and stability.

## A.2 OFF-POLICY INFORMATION CONSTRUCTION WITH INFOSEEK DATASET

As introduced in Section 2.3, our process-based reinforcement learning approach relies on densely supervised data. This appendix details how we construct this off-policy supervision, specifically the weighted sub-goals and hints, by leveraging the unique structure of the **InfoSeek** dataset (Xia et al., 2025). Figure 5, 6 and 7 provide three examples.

**InfoSeek: A Dataset Built on Decomposable Reasoning Structures.** The InfoSeek dataset was specifically designed to address the scarcity of benchmarks for *Deep Research* tasks, which demand complex, multi-step reasoning beyond simple multi-hop question answering. Its core innovation lies in its data synthesis paradigm, which generates questions grounded in a verifiable and explicit reasoning structure called a **Research Tree**. The generation process begins by mining entities and their relationships from a large-scale text corpus. From these, a "Research Tree" is recursively constructed for each data point, where the root denotes the final, unique answer, internal nodes represent intermediate sub-goals, and edges encode their logical dependencies. To ensure complexity, the descriptions of these internal nodes are "blurred" with additional constraints. Finally, a powerful LLM is prompted with the entire tree structure to generate a high-level, natural language question whose resolution requires traversing the entire reasoning path. This tree-based structure provides a

ground-truth decomposition of a complex problem into a hierarchy of verifiable sub-goals, making it an ideal foundation for generating process-level supervision.

**InfoSeek-Evaluation**   The InfoSeek-Evaluation set contains 300 high-quality, human-checked samples to evaluate agentic deep search capability. Qwen2.5-72B-Instruct with a CoT prompting achieves lower than 8% accuracy in this evaluation set.

**Constructing Weighted Sub-Goals.**   We utilize the InfoSeek Research Tree's topology to define sub-goals and assign an importance weight $s_i$ to each. Our process begins by extracting a subset of **high-value** internal nodes from the Research Tree to form the set $\mathcal{G}_q = \{g_1, \ldots, g_M\}$, deliberately excluding simple confirmatory facts. We leverage a powerful teacher model, Gemini 2.5 Pro, to meticulously select these critical entities (typically 2-4 per tree) and assign an importance weight to each. This selection process distinguishes between **pivotal intermediate nodes** (core entities unlocking subsequent paths) and **secondary supporting nodes** (necessary evidence), ensuring sparse yet targeted supervision. The specific prompt used for this task is detailed in Appendix A.6. The assigned weights are constrained to sum to one ($\sum_{i=1}^{M} s_i = 1$), providing the final set of weighted sub-goals $\{(g_i, s_i)\}_{i=1}^{M}$ for our reward shaping scheme.

**Generating Hints as Guiding Queries.**   Hints are formulated as high-leverage guiding queries that act as off-policy information bridges. They are designed to assist the agent when it is unable to make progress through autonomous exploration, thereby mitigating unproductive reasoning loops. These hints are generated using Gemini 2.5 Pro (see Appendix A.6) based on the critical edges of the Research Tree. During policy optimization, these hints are instrumental in teaching the agent several crucial search skills. They foster **purposeful search** by providing direct queries for specific sub-problems, guiding the agent onto a productive path. Furthermore, they help the agent **break through key points** in the reasoning chain where identifying the next step is non-obvious. Finally, by reframing or combining constraints in novel ways, the hints encourage **creative search**, training the agent to formulate more effective queries beyond simple keyword matching.

Figure 5, 6 and 7 provide three examples. The main question contains multiple, intertwined constraints. The generated hints effectively decompose this complexity by isolating and combining key constraints into actionable search queries. The first hint focuses on identifying the person, while the second provides an alternative, more robust query by combining the person's profession with their marital information.

---

**Question:** What is a literary genre that was defined by a novelist who wrote a novel incorporating elements of the legendary origins of the Hope Diamond, and was mentored by Charles Dickens, characterized as a 'novel-with-a-secret'?

---

**Answer:** Sensation novel

---

**Hint Queries:**
  *novelist mentored by Charles Dickens who wrote The Moonstone*

  *author whose novel incorporated elements of the Hope Diamond and was mentored by Charles Dickens*

  *author of 'The Woman in White' mentored by Charles Dickens*

---

**Sub Goals:**
Wilkie Collins: weight 0.6
Charles Dickens: weight 0.2
The Moonstone: weight 0.2

---

Figure 5: Case study 1 (Sensation novel): An example of enriched InfoSeek dataset. The hints decompose the main question into more manageable, high-leverage search queries that serve as off-policy guidance.

Through this process, we enrich the original InfoSeek dataset with a structured layer of off-policy supervision. This augmented data, containing both quantitative sub-goal importance and qualitative reasoning hints, provides a robust foundation for training more capable and efficient Deep Research agents using our proposed reinforcement learning framework.

**Question:** What is an album that was created by a musician who played piano in Gus Arnheim's band, created a jazz camp, was recorded in 1955, and features drumming by Mel Lewis?

**Answer:** Contemporary Concepts

**Hint Queries:**

*musician who played piano in Gus Arnheim's band and later created a jazz camp*

*bandleader whose 1955 album featured Mel Lewis on drums*

*jazz pianist who once played for Gus Arnheim and founded a music education program*

**Sub Goals:**
Stan Kenton: weight 0.7
Gus Arnheim: weight 0.3

Figure 6: Case study 2 (Contemporary Concepts): An example of enriched InfoSeek dataset. The hints decompose the main question into more manageable, high-leverage search queries that serve as off-policy guidance.

**Question:** What is a British Thoroughbred racehorse that was sired by a horse who won the 1941 Epsom Derby, was the leading British two-year-old of 1959, was a dark bay horse with a white blaze standing 16.1 hands high, and had considerable success as a sire of sprinters?

**Answer:** Sing Sing (horse)

**Hint Queries:**

*horse that won the 1941 Epsom Derby*

*1941 Epsom Derby winner*

**Sub Goals:**
Tudor Minstrel: weight 0.5
Owen Tudor: weight 0.5

Figure 7: Case study 3 (Sing Sing (horse)): An example of enriched InfoSeek dataset. The hints decompose the main question into more manageable, high-leverage search queries that serve as off-policy guidance.

## A.3 FURTHER DUAL-AGENT FRAMEWORK EXPERIMENTS AND EFFICIENCY ANALYSIS

As introduced in Section 3.1, our dual-agent framework decouples high-level reasoning from low-level evidence gathering to enhance performance and efficiency. This section provides a detailed empirical analysis of this design.

Table 5: Analysis of the dual-agent framework on the InfoSeek evaluation set. The Researcher Agent is fixed as Qwen2.5-3B-Instruct. "Context Length" is the average number of tokens processed by the researcher per trajectory. "Time" denotes the average inference time per task.

| Refiner Agent | Accuracy (%) | Search Calls (#) | Context Length (Tok.) | Time (min.) |
|---|---|---|---|---|
| w/o refiner | 8.4 | 1.93 | 2372.4 | 12.2 |
| Qwen2.5-3B-Inst | 13.4 | 3.07 | 1309.6 | 10.2 |
| Qwen2.5-7B-Inst | 17.2 | 2.83 | 1071.2 | 10.5 |
| Qwen2.5-32B-Inst | 18.8 | 3.01 | 1260.4 | 11.3 |

As shown in Tab 5, we conduct analytical study employing a fixed Qwen2.5-3B-Instruct researcher to isolate the impact of the refiner with InfoSeek evaluation set. The baseline without a refiner struggles, achieving only 7.4% accuracy. The introduction of a 3B refiner dramatically improves accuracy to 13.4% while simultaneously reducing the researcher's average context length per trajectory by 45% (from 2372 to 1310 tokens). Scaling the refiner to a 7B model yields further gains to 17.2%. This demonstrates that offloading evidence distillation enables the researcher to dedicate their limited context window to high-level reasoning, significantly boosting performance.

Beyond performance gains, the dual-agent framework offers computational efficiency. The primary bottleneck in LLM is the quadratic complexity ($O(n^2)$) of self-attention with respect to context length. By delegating the processing of verbose evidence to the refiner, we substantially reduce the peak context length for the researcher. In a practical deployment, this architecture is highly feasible. A standard setup for information-seeking tasks already requires a researcher agent and a retrieval service ( 10% VRAM) in a single 8xH800 node. Adding a dedicated refiner, optimized with frameworks like vLLM, incurs a manageable overhead of approximately 20% more VRAM, making the entire system viable on a single 8xH800 node.

A key advantage of our approach is its implementation simplicity and adaptability. Unlike complex multi-agent reinforcement learning schemes, our refiner can be aligned with the researcher via a straightforward SFT process. This involves sampling trajectories from the researcher and using them to train the refiner, ensuring it learns to distill information in a manner tailored to the researcher's reasoning patterns. Consequently, the refiner is not a static, prompt-engineered module but a dynamic component that co-evolves with the researcher. This training methodology provides a scalable path toward building more capable, collaborative agent systems without incurring prohibitive complexity.

### A.4 IMPLEMENTATION DETAILS

For research agent RFT, we fine-tune for 3 epochs with a learning rate of 1e-5, L2 normalization of 0.01(important for stablizing training), and a context length of 16,384, using a single 8×H100 node. For refiner agent RFT, we fine-tune for 2 epochs with a learning rate of 1e-5, L2 normalization of 0.01 , and a context length of 8,192, using a single 8×H100 node.

RL training is conducted with a batch size of 256, a maximum of 10 turns, rollout size 8, temperature 0.8, and a search engine restricted to the top-5 retrieved contents. The training is conducted on two 8×H100 nodes.

### A.5 THE USE OF LARGE LANGUAGE MODELS (LLMS)

LLMs are used to polish writing and are used for enriching the training dataset, which is described in Sec 2.3.

## A.6 PROMPTS

### A.6.1 REFINER AGENT

The complete prompt template used for our Refiner Agent is presented in Listing 1. Note that we use this template both to drive the refiner and for refiner RFT training

```
<|im_start|>user
**TASK:**
Synthesize the key information from the **[Retrieved Documents]** that
    ↪ is relevant to the **[Current Query]**. The synthesis should be
    ↪ guided by conducting deep research to uncover the **[Original
    ↪ Question]**.

**INSTRUCTIONS:**
1.  **Extract & Merge:** Identify all relevant facts and combine them.
    ↪ Eliminate redundancy. You should provide information for deep
    ↪ research, not answer to current query or original question.
2.  **Provide Information, Not an Answer:** Your output should be a
    ↪ self-contained block of information, NOT a direct, short answer
    ↪ to the original question or the current query.
3.  **Handle Insufficient Information:** If the documents do not
    ↪ contain relevant information for the query, state that the
    ↪ provided sources are insufficient and suggest that further
    ↪ investigation may be needed. You can also provide some further
    ↪ investigation direction and query rewrite suggestions.
4.  **Format:** Enclose the entire synthesized output within `<
    ↪ information>` and `</information>` tags. Add no other text. For
    ↪ example, <information> Synthesized information for deep research
    ↪ here </information>.

**CONTEXT:**
- **[Original Question]:** {original_question}
- **[Current Query]:** {query}
- **[Retrieved Documents]:** {documents}

**TASK:**
Synthesize the key information from the **[Retrieved Documents]** that
    ↪ is relevant to the **[Current Query]**. The synthesis should be
    ↪ guided by conducting deep research to uncover the **[Original
    ↪ Question]**.

**INSTRUCTIONS:**
1.  **Extract & Merge:** Identify all relevant facts and combine them.
    ↪ Eliminate redundancy. You should provide information for deep
    ↪ research, not answer to current query or original question.
2.  **Provide Information, Not an Answer:** Your output should be a
    ↪ self-contained block of information, NOT a direct, short answer
    ↪ to the original question or the current query.
3.  **Handle Insufficient Information:** If the documents do not
    ↪ contain relevant information for the query, state that the
    ↪ provided sources are insufficient and suggest that further
    ↪ investigation may be needed. You can also provide some further
    ↪ investigation direction and query rewrite suggestions.
4.  **Format:** Enclose the entire synthesized output within `<
    ↪ information>` and `</information>` tags. Add no other text. For
    ↪ example, <information> Synthesized information for deep research
    ↪ here </information>.

**SYNTHESIZED INFORMATION:**
<|im_end|>
<|im_start|>assistant
```

Listing 1: The prompt template for the Refiner Agent.

### A.6.2 PROMPT FOR DATASET ENRICHMENT

We use the Gemini 2.5 API (Gemini Team, 2025) with the following prompt to conduct InfoSeek dataset enrichment as described in Section 2.3 and Section A.2.

```
<|im_start|>user
**Role**: You are an AI Data Augmentation expert. Your mission is to
    ↪ extract and expand key information from a Research Tree to
    ↪ optimize reinforcement learning for training an LLM as a deep
    ↪ research agent.

**Objective**: From the input Research Tree, complete the two tasks
    ↪ below and return results in one unified JSON output.

### **Task 1: Extract High-Value Entities & Assign Weights (for Reward
    ↪ Shaping)**

Identify pivotal breakthroughs to reward in PPO training.

**Steps**:
1. Select **2-4 most critical entities** from the Research Tree.
2. Assign each a `weight` (float), with all weights summing to **1.0**.
3. Prioritize:
    * **Pivotal Nodes (0.6-0.8)**: Core breakthroughs, usually direct
        ↪ children of the root, resolving major clauses.
    * **Supporting Nodes (0.2-0.4)**: Necessary for pivotal nodes,
        ↪ smaller but still important.
    * Exclude trivial confirmatory facts.

**Output**: JSON array of objects with `id`, `entity`, and `weight`.

### **Task 2: Generate Early-Stage Guiding Queries (for Strategic Hints
    ↪ )**
Provide hints to guide initial exploration without leaking answers.

**Steps**:
1. Generate **1-2 critical guiding queries**.
2. Focus on **leaf nodes**, using their parent's entity + claim.
3. Queries must **not** contain the child node's entity.
4. Queries should be natural, strategic, and yield high information
    ↪ gain.

**Output**: JSON array of objects with `target_id` and `
    ↪ generated_queries` (array of strings).

**Background**:
* Research Tree = hierarchical structure of questions/answers (nodes).
* Root = original complex question.
* Children = sub-questions.
* Claims = relationship between parent and child entities.

**Example Input & Output:**
...

**Execute both tasks on this Research Tree:**
{research_tree_stucture}

**Output:**
<|im_end|>
<|im_start|>assistant
```

Listing 2: The prompt for the AI Data Augmentation expert to process the Research Tree.

