# OpenReview forum: "Reinforcing Agentic Search Via Reward Density Optimization"
_ICLR.cc/2026/Conference — ICLR 2026 Conference Withdrawn Submission_

### Official Review · Reviewer_a4QG · 2025-10-26

**Soundness:** 3
**Presentation:** 2
**Contribution:** 3
**Rating:** 4
**Confidence:** 2

**Summary:**

This paper proposes a dual-agent mechanism for Deep Search Question Answering (DSQA) using LLMs. The method makes three main contributions: (1) a dual-agent framework with a researcher agent that plans and takes actions and a refiner agent that summarizes gathered information; (2) the introduction of subgoals to guide the search; and (3) off-policy hints to improve agent performance.

**Strengths:**

1- Comparison to strong recent baselines.
2- A wide range of datasets.
3- Outperforming all baselines in most of the datasets.

**Weaknesses:**

1- The presentation should be much clearer. For example, the process for generating hints needs to be described in sufficient detail in the main paper, with a pointer to the appendix for further specifics. Currently, the main paper provides no explanation of how the off-policy hints are produced.

2- Although the proposed method clearly outperforms the baselines, the performance gains seem trivial. If we effectively double the resources by employing two agents simultaneously and further add subgoals and hints, it is expected that the method would surpass agents that lack these tools. Moreover, the hints are generated by a separate LLM (Gemini 2.5 Pro), indicating that the approach relies on even more resources.

**Questions:**

1- How are the off-policy hints generated? Are they constructed manually?

2- The paper states that each node in the reasoning tree is a subgoal to achieve. This would include nodes that are not on the final path. Is treating all such nodes as subgoals/milestones the intended and correct approach?

3- Could you clarify the statement: “Following the official implementation, accuracy is judged by an LLM.” What exactly does “judged by an LLM” entail?

4- In Table 2, when comparing models of the same size, your method employs two LLM agents. Does this effectively mean your approach uses roughly twice the parameters/resources of the other baselines?

---

> ### Author Response · Authors · 2025-11-25
> **Response to Reviewer a4QG (1)**
>
> > W1: The presentation should be much clearer. For example, the process for generating hints needs to be described in sufficient detail in the main paper, with a pointer to the appendix for further specifics. Currently, the main paper provides no explanation of how the off-policy hints are produced.
> * Thank you for the comment. We agree that the description of hint generation should be clearer. In the revised version, we have added a dedicated paragraph in the main paper explaining how off-policy hints are generated (Sec. 2.3). Additional prompt formats and examples are provided in the appendix. This resolves the earlier lack of clarity.
>
> > W2: Although the proposed method clearly outperforms the baselines, the performance gains seem trivial. If we effectively double the resources by employing two agents simultaneously and further add subgoals and hints, it is expected that the method would surpass agents that lack these tools. Moreover, the hints are generated by a separate LLM (Gemini 2.5 Pro), indicating that the approach relies on even more resources.
>
> Thank you for the question. We would like to clarify that the performance gains are **not** merely a result of doubling compute or relying on a stronger external LLM. Our main contribution concerns *reward density optimization*, a challenge that persists regardless of model size or agent count. The key purpose of the dual-agent framework and the process-level supervision is to **stabilize long-horizon exploration**, not to inflate resource usage.
> * **1. Dual-agent does not double inference time cost.** As analyzed in Appendix A.3, when deployed under the *same* compute budget (8×H100), the dual-agent setup is actually **faster** than the single-agent baseline (10.2 min vs. 12.2 min). This is because the dual agent prevents context accumulation, reducing quadratic attention cost.
> * **2. Sub-goals and hints are not teacher model driven performance boosters.** These components are introduced to address **low reward density**, which remains the central bottleneck in RL-based deep search. Their function is to **encourage and guide the model to persistently engage in long-horizon exploration**, not to add external signal strength.
> * To demonstrate this, we additionally induced sub-goals and hints on **HotpotQA**, a dataset without explicit structure, using **Qwen3-8B**, a standard open-source model. Even with this modest annotator, we observe consistent improvements:
>
> | Training Set and Method | NQ   | TQ   | PopQA | HotpotQA | 2Wiki | Musique | Bamboogle | AVG      |
> | ----------------------- | ---- | ---- | ----- | -------- | ----- | ------- | --------- | -------- |
> | HQA                     | 40.7 | 59.6 | 44.4  | 45.1     | 50.4  | 19.1    | 46.9      | 43.7     |
> | **HQA + SubGoal&Hints** | 42.5 | 60.9 | 47.3  | 48.7     | 53.5  | 19.7    | 44.8      | **45.3** |
>
> The **+1.6 average gain** shows that the benefit persists even when the supervision is produced by a small, publicly available model, i.e., the method does *not* rely on high-end APIs.
>
> > Q1: How are the off-policy hints generated? Are they constructed manually?
> * The off-policy hints are not constructed manually. They are automatically generated using LLMs (e.g., Gemini 2.5 or Qwen-3-8B). Specifically, the process involves extracting critical edges from the ground-truth reasoning tree and prompting the LLM to transform them into high-leverage guiding queries. These queries are designed to decompose complex constraints into actionable search steps (e.g., purposeful search or constraint reframing).
> * We have revised Section 2.3 in the main paper to explicitly describe this generation pipeline and the specific skills these hints target.

---

> ### Author Response · Authors · 2025-11-25
> **Response to Reviewer a4QG (2)**
>
> > Q2: The paper states that each node in the reasoning tree is a subgoal to achieve. This would include nodes that are not on the final path. Is treating all such nodes as subgoals/milestones the intended and correct approach?
> * Thank you for the question. Our intention is not to treat every node in the reasoning tree as a rewarded sub-goal. To avoid reward hacking, we use an LLM annotator to select only the **most important nodes**, i.e., milestones necessary for solving the full problem. Only these selected nodes receive sub-goal reward. This ensures dense but targeted supervision rather than rewarding every branch in the tree.
>
> > Q3: Could you clarify the statement: “Following the official implementation, accuracy is judged by an LLM.” What exactly does “judged by an LLM” entail?
> * We follow the official evaluation protocol: correctness is judged by an LLM. Concretely, we extract the span enclosed in `<answer>...</answer>` and use **DeepSeek-v3.1** to compare the predicted answer against the gold answer. Since the final answer is short and structured, even smaller models can reliably evaluate correctness, but we adopt DeepSeek-v3.1 to eliminate borderline misclassification.
>
> > Q4: In Table 2, when comparing models of the same size, your method employs two LLM agents. Does this effectively mean your approach uses roughly twice the parameters/resources of the other baselines?
>
> This is addressed quantitatively in **Appendix A.3**. Under identical compute (8×H100), we compare the inference-time cost of a single-agent baseline and our dual-agent framework.
>
> Because a single agent must process an ever-growing context, its attention cost increases quadratically with sequence length. The dual-agent design avoids this by offloading and compressing evidence, preventing context bloat. As a result:
>
> * **Single agent:** 12.2 minutes (per evaluation)
> * **Dual agent:** **10.2 minutes**
>
> Thus, even with two agents, inference is **faster**. This reflects a general property of dual-agent workflows, and the comparison is representative of conventional baselines such as Search-R1, AutoRefine, and InForage, which all operate in the single-agent accumulation setting.

---

> ### Author Response · Authors · 2025-11-28
>
> Dear Reviewer a4QG,
>
> Thank you once again for your valuable comments on our submission. As the discussion phase is approaching its end, we would like to kindly confirm whether we have sufficiently addressed all of your concerns (or at least part of them). Should there be any remaining questions or areas requiring further clarification, please do not hesitate to let us know. If you are satisfied with our responses, we would greatly appreciate your consideration in adjusting the evaluation scores accordingly.
>
> We sincerely look forward to your feedback.

---

### Official Review · Reviewer_hDmp · 2025-10-31

**Soundness:** 3
**Presentation:** 3
**Contribution:** 3
**Rating:** 6
**Confidence:** 4

**Summary:**

This paper introduces InfoFlow, a dual-agent framework designed to address the problem of low reward density when training language model agents for agentic deep search tasks. The framework tackles this by combining three techniques: (1) decomposing tasks into sub-problems and shaping rewards at intermediate stages, (2) using adaptive off-policy hints to guide exploration and rescue failed rollouts, and (3) introducing a division of labor between a “researcher” agent (responsible for high-level planning) and a “refiner” agent (summarizing retrieved evidence to reduce the trajectory context burden). InfoFlow is empirically evaluated on a series of benchmarks for deep question answering and agentic search.

**Strengths:**

1.   The paper addresses a well-motivated and topical challenge in RL for information-seeking agents: the problem of sparse and low-density rewards in multi-step reasoning and deep search. The reward shaping method, which leverages explicit subgoal decomposition with grounded, weighted milestones, is conceptually clear and technically well-integrated.
2.   InfoFlow’s architecture is thoughtfully designed: separating researcher and refiner agents is a practical step toward reducing context length and making deeper reasoning more feasible. Figure 2 compellingly visualizes this effect, showing accuracy gains along with drastic context length reduction, which directly supports the claim about memory bottlenecks in monolithic LLM agents.
3.   The use of adaptive off-policy hints is well-motivated and addresses a real limitation in on-policy RL for sparse, long-horizon tasks.

**Weaknesses:**

1.   The method’s tight coupling to the InfoSeek dataset (with its tree-structured subgoal annotation and LLM-generated hints) limits broader impact. While the technique is shown to outperform prior work on standard QA benchmarks, the core innovation (dense, process-based guidance) seems most advantageous only when such decomposition is available.
2.   There are no experiments analyzing the quality, coverage, or utilization/impact of hints injected during RL—an important aspect given the claimed centrality of off-policy hints to exploration. Similarly, the comparative breakdown of successful interventions via hint-injection (at various values of $K_h$ or types of hints) is missing.

**Questions:**

1.   Could the impact of increased computation or inference cost in the dual-agent framework be quantified relative to conventional approaches?
2.   In Table 1, the “solve none” rate remains elevated (>45%) even after co-training, indicating that difficult instances remain largely unsolved. What are the key failure modes, and do they cluster by reasoning depth or other properties?

---

> ### Author Response · Authors · 2025-11-25
> **Response to Reviewer hDmp (1)**
>
> > W1: The method’s tight coupling to the InfoSeek dataset (with its tree-structured subgoal annotation and LLM-generated hints) limits broader impact. While the technique is shown to outperform prior work on standard QA benchmarks, the core innovation (dense, process-based guidance) seems most advantageous only when such decomposition is available.
>
> Thank you for the insightful comment. We would like to clarify that our method is **not** designed to rely on the specific tree annotations of InfoSeek, nor to achieve gains simply by using more complex supervision. Instead, our central claim is that **reward density is a fundamental bottleneck in RL-based deep search**, and that *retaining high-value intermediate information* (e.g., sub-goal structure and failure-guided hints) is crucial for stabilizing optimization in future deep-search agents.
>
> * **Why this is not coupling to InfoSeek training dataset.** Deep search tasks naturally exhibit a **hierarchical structure**: horizontally, multiple constraints must be satisfied before advancing; vertically, multi-hop reasoning is required to reach the final answer. Such structure exists independently of InfoSeek and can be extracted or approximated in other domains. Our framework exploits this general property to provide denser, process-based signals; it does not assume any dataset-specific ontology.
> * **Evidence of transfer beyond InfoSeek.** To demonstrate that the approach is *not* bound to InfoSeek’s tree annotations, we constructed sub-goals and hints on **HotpotQA**, a dataset that does not natively provide such decomposition. Using Qwen3-8B to enrich annotations, we compare HQA baselines with/without sub-goal & hint supervision:
>
> | Training Set and Method | NQ   | TQ   | PopQA | HotpotQA | 2Wiki | Musique | Bamboogle | AVG      |
> | ----------------------- | ---- | ---- | ----- | -------- | ----- | ------- | --------- | -------- |
> | InfoSeek + SubGoal&Hints| 47.2 | 68.1 | 48.1  | 44.3     | 47.2  | 21.9    | 47.6      | 46.2     |
> | HQA                     | 40.7 | 59.6 | 44.4  | 45.1     | 50.4  | 19.1    | 46.9      | 43.7     |
> | **HQA + SubGoal&Hints** | 42.5 | 60.9 | 47.3  | 48.7     | 53.5  | 19.7    | 44.8      | 45.3     |
>
> Adding sub-goals and hints to HQA improves the *average* by **+1.6** across seven QA-based search tasks. Importantly, this improvement arises even though HotpotQA does not contain explicit reasoning trees, showing that our method is **dataset-agnostic** and benefits any task where approximate sub-problem structure can be induced.
>
> * **Further analysis.** Our intention is not to propose a dataset-specific technique, but rather to show that **reward density optimization requires preserving intermediate, verifiable supervision signals** during data construction or synthesis. InfoSeek merely provides one clean instantiation of such signals; the framework itself is general-purpose and applicable wherever tasks have decomposable structure (e.g., multi-hop QA, web-agent benchmarks, long-horizon tool-use).
>
> > W2: There are no experiments analyzing the quality, coverage, or utilization/impact of hints injected during RL—an important aspect given the claimed centrality of off-policy hints to exploration. Similarly, the comparative breakdown of successful interventions via hint-injection (at various values of or types of hints) is missing.
>
> Thank you for pointing out the need for a more explicit analysis of hint quality and impact. We have added two sets of experiments to address this.
>
> * **Direct evaluation of hint effectiveness.** We conducted an inference-time study on 500 held-out InfoSeek samples, comparing models initialized with dual-agent RFT *with* vs. *without* hint injection. The hints used here are exactly the guiding queries constructed during training.
>
> | Method        | Mean@4 |
> | ------------- | ------ |
> | RFT3B-NoHints | 33.7   |
> | RFT3B-Hints   | 37.9   |
>
> This shows that hints improve success rate even *before* RL training. Moreover, as RL progresses, the agent learns to exploit these guided search patterns more effectively, consistent with our claim that hints accelerate early exploration rather than acting as a crutch.
>
> * **RL training dynamics confirm that hints yield meaningful gains rather than superficial reward increases.** In the **revision (Sec. 4.3.3)**, we include training curves comparing models with and without hint injection. We report both the *original final reward* and the *full shaped reward* (sub-goal + final). Both curves improve jointly when hints are used. This indicates that hints do not inflate the shaped reward artificially; instead, they increase the rate of successful trajectories, which in turn raises the actual task reward.

---

> ### Author Response · Authors · 2025-11-25
> **Response to Reviewer hDmp (2)**
>
> > Q1: Could the impact of increased computation or inference cost in the dual-agent framework be quantified relative to conventional approaches?
> Yes. We provide a quantitative comparison in **Appendix A.3**, where we evaluate inference-time cost under identical compute (8×H100) for a **single-agent** setup versus our **dual-agent** framework.
>
> Because the single agent must handle a much longer accumulated context, its attention cost grows quadratically with context length. In contrast, the dual-agent framework continuously offloads and compresses retrieved evidence, preventing context bloat. As a result, **dual-agent inference is faster despite involving two models**:
>
> * **Single agent:** 12.2 minutes (per evaluation)
> * **Dual agent:** **10.2 minutes**
>
> This speedup reflects a general property of the dual-agent architecture, not an artifact of a specific implementation. Conventional approaches such as Search-R1, AutoRefine, or InForage follow the same “single-agent accumulating context” paradigm (aside from prompt format and optimization strategy), so the comparison is representative.
>
> > Q2: In Table 1, the “solve none” rate remains elevated (>45%) even after co-training, indicating that difficult instances remain largely unsolved. What are the key failure modes, and do they cluster by reasoning depth or other properties?
>
> Thank you for raising this point. The high “solve-none” rate reflects the inherent difficulty of the hardest DSQA cases rather than instability of the method. Following your suggestion, we analyzed a subset of failed trajectories from **BrowseComp-Plus** and **InfoSeek-Eval** to understand the dominant failure modes:
>
> * **(1) Early decomposition and planning errors (18.1%).** A common source of failure is incorrect initial decomposition: the agent commits to a suboptimal search direction in the first few turns and fails to identify a viable entry point into the reasoning graph. Once the early plan drifts, later steps provide little meaningful information, and the trajectory never recovers.
> * **(2) Incomplete long-horizon reasoning (33.8%).** We observe cases where the agent forms a partially correct intermediate hypothesis but answers prematurely rather than fully validating all constraints. This reflects the inherent difficulty of sustaining multi-step search and verification over long horizons.
> * **(3) Retrieval errors (48.2%).** Nearly half of the failures stem from missing or low-quality evidence: the issued sub-queries fail to surface the needed document, or relevant pages are ranked too low by the retriever. Even when the reasoning direction is correct, repeated query rewrites often fail to recover the target information.
>
> These findings clarify why many difficult examples remain unsolved and highlight retrieval fidelity and long-horizon consistency as key directions for further improvement.

---

> ### Author Response · Authors · 2025-11-28
>
> Dear Reviewer hDmp,
>
> Thank you once again for your valuable comments on our submission. As the discussion phase is approaching its end, we would like to kindly confirm whether we have sufficiently addressed all of your concerns (or at least part of them). Should there be any remaining questions or areas requiring further clarification, please do not hesitate to let us know.
>
> We sincerely look forward to your feedback.

---

### Official Review · Reviewer_weiU · 2025-10-31

**Soundness:** 2
**Presentation:** 3
**Contribution:** 2
**Rating:** 2
**Confidence:** 4

**Summary:**

The paper targets deep-search agent tasks and proposes a pipeline named InfoFlow to tackle the sparse-reward problem in RLVR settings. The method introduces sub-goal–based intermediate rewards, injects hints into partially failed trajectories to increase success probability, and employs a refiner agent to summarize long texts and generate concise representations.

**Strengths:**

1. The paper addresses an important and common challenge in deep-search agent tasks.
2. The writing is clear and easy to follow.
3. The use of an auxiliary refiner agent to summarize and compress task information is interesting and potentially generalizable.

**Weaknesses:**

1. The main innovations claimed by this paper: sub-goal–based dense rewards and failure-guided hints are primarily data-driven enhancements rather than methodological advances. Both rely on stronger external LLM APIs for sub-goal generation and hint generation through specifically designed prompts. This makes the approach highly engineered and dataset-specific rather than an automated, generalizable "strategy". The contribution is therefore weakened, as the benefit largely stems from using denser supervision from stronger models. In contrast, many prior works have proposed principled, model-independent methods for reward densification, from general reward shaping in RL to process reward models in LLM training. Thus, relying on the LLM API to provide a sub-goal reward applicable only to customized datasets seems highly intensive and lacks innovation, and should certainly not be considered the main contribution of the paper.
2. Since the sub-goal decomposition, sub-goal reward, and trajectory hints are all derived from external APIs, the method distills additional knowledge into the training data. Comparing this enhanced dataset against baselines trained on unaugmented data is unfair and does not demonstrate algorithmic superiority, only data quality differences.
3. In Section 3.2.1, the use of the term *rejection sampling* is inaccurate, as the described process is a deterministic data filtering procedure rather than probabilistic rejection sampling in the Monte Carlo sense. I suggest the authors revise this terminology to avoid misunderstanding.
4. The paper provided an anonymous code link, but the repository is empty (at least at the time of review).

**Questions:**

See the weakness

---

> ### Author Response · Authors · 2025-11-25
> **Response to Reviewer weiU (1)**
>
> > W1: The main innovations claimed by this paper: sub-goal–based dense rewards and failure-guided hints are primarily data-driven enhancements rather than methodological advances. Both rely on stronger external LLM APIs for sub-goal generation and hint generation through specifically designed prompts. This makes the approach highly engineered and dataset-specific rather than an automated, generalizable "strategy".  The contribution is therefore weakened, as the benefit largely stems from using denser supervision from stronger models. In contrast, many prior works have proposed principled, model-independent methods for reward densification, from general reward shaping in RL to process reward models in LLM training. Thus, relying on the LLM API to provide a sub-goal reward applicable only to customized datasets seems highly intensive and lacks innovation, and should certainly not be considered the main contribution of the paper.
> > W2: Since the sub-goal decomposition, sub-goal reward, and trajectory hints are all derived from external APIs, the method distills additional knowledge into the training data. Comparing this enhanced dataset against baselines trained on unaugmented data is unfair and does not demonstrate algorithmic superiority, only data quality differences.
>
> We would like to clarify that the goal of our work is **not** to rely on a customized dataset or to buy gains from stronger external APIs. Our central claim is that **reward density is a fundamental bottleneck in RL-based deep search**, and that *retaining high-value intermediate information* (such as sub-goal structure and failure-guided hints) is essential for enabling stable long-horizon optimization. The essence behind these components is to **encourage and guide the model to persistently engage in long-horizon exploration**, rather than to engineer dataset-specific advantages.
> * **1. Why this is not coupling to the InfoSeek dataset.** Deep search tasks (regardless of dataset) exhibit inherent **hierarchical structure**: horizontally, multiple constraints must be jointly satisfied at each stage; vertically, multi-hop reasoning is required before the final answer becomes reachable. Our method leverages this general property to provide denser, process-level signals. It does *not* assume the existence of a particular tree format or dataset-specific ontology, and the framework applies whenever such structure can be approximated or induced.
> * **2. Evidence that the method transfers beyond InfoSeek and beyond stronger external LLM APIs.** To show that our approach is not bound to InfoSeek’s annotations or any proprietary model, we reconstructed sub-goals and hints on **HotpotQA**, which does *not* contain explicit decomposition. Using **Qwen3-8B** (a publicly available model) to annotate intermediate structure, we trained agents with/without sub-goal and hint supervision and evaluated on seven QA-based search tasks:
>
> | Training Set and Method  | NQ   | TQ   | PopQA | HotpotQA | 2Wiki | Musique | Bamboogle | AVG      |
> | ------------------------ | ---- | ---- | ----- | -------- | ----- | ------- | --------- | -------- |
> | InfoSeek + SubGoal&Hints | 47.2 | 68.1 | 48.1  | 44.3     | 47.2  | 21.9    | 47.6      | 46.2     |
> | HQA                      | 40.7 | 59.6 | 44.4  | 45.1     | 50.4  | 19.1    | 46.9      | 43.7     |
> | **HQA + SubGoal&Hints**  | 42.5 | 60.9 | 47.3  | 48.7     | 53.5  | 19.7    | 44.8      | **45.3** |
>
> Adding sub-goals and hints to HQA yields a **+1.6** improvement on average. This demonstrates that (1) the approach is **dataset-agnostic**, and (2) meaningful sub-goal structures can be induced with small-scale open LLMs, strong external APIs are not a prerequisite.

---

> ### Author Response · Authors · 2025-11-25
> **Response to Reviewer weiU (2)**
>
> > W3: In Section 3.2.1, the use of the term rejection sampling is inaccurate, as the described process is a deterministic data filtering procedure rather than probabilistic rejection sampling in the Monte Carlo sense. I suggest the authors revise this terminology to avoid misunderstanding.
>
> Thank you for pointing this out. We agree that the term “rejection sampling’’ can be overloaded across communities. In RL and Monte Carlo literature it typically denotes a probabilistic procedure, while in recent LLM-reasoning and agentic-search work, *the same term has been widely used to refer to sampling multiple trajectories and keeping only the correct ones*. Examples include:
>
> * *A minimalist approach to LLM reasoning: from rejection sampling to REINFORCE*
> * *Scaling Relationship on Learning Mathematical Reasoning with LLMs*
> * *WebDancer: Towards Autonomous Information-Seeking Agency* (NeurIPS 2025)
>
> Our usage follows this emerging convention in LLM training pipelines, where “rejection sampling’’ denotes *multi-sample trajectory generation followed by filtering for correctness*. To avoid confusion, we will revise the text to explicitly state this interpretation and clarify that the procedure is deterministic filtering over sampled trajectories rather than Monte Carlo rejection sampling in the classical sense.
>
> [1]: A minimalist approach to llm reasoning: from rejection sampling to reinforce
>
> [2]: Scaling Relationship on Learning Mathematical Reasoning with Large Language Models
>
> [3]: Webdancer: Towards autonomous information seeking agency (Accepted by NeurIPS 2025)
>
> > W4: The paper provided an anonymous code link, but the repository is empty (at least at the time of review).
>
> We apologize for the confusion. When updating the main GitHub repository, we inadvertently forgot to synchronize the anonymous mirror. The anonymous repository has now been fully updated with the complete code and data-processing scripts.

---

> ### Author Response · Authors · 2025-11-28
>
> Dear Reviewer weiU,
>
> Thank you once again for your valuable comments on our submission. As the discussion phase is approaching its end, we would like to kindly confirm whether we have sufficiently addressed all of your concerns (or at least part of them). Should there be any remaining questions or areas requiring further clarification, please do not hesitate to let us know. If you are satisfied with our responses, we would greatly appreciate your consideration in adjusting the evaluation scores accordingly.
>
> We sincerely look forward to your feedback.

---

### Official Review · Reviewer_GJJo · 2025-10-31

**Soundness:** 3
**Presentation:** 3
**Contribution:** 3
**Rating:** 6
**Confidence:** 3

**Summary:**

This paper proposes InfoFlow, a reinforcement-learning framework for agentic deep search that tackles sparse rewards by optimizing reward density through three ingredients: (i) sub-goal reward shaping from a research-tree decomposition with importance-weighted sub-goals, (ii) adaptive off-policy hints that inject high-leverage guiding queries when rollouts stall, and (iii) a dual-agent architecture where a Refiner condenses retrieved evidence for a Researcher to stabilize and shorten reasoning–search loops; training is bootstrapped via rejection-sampling fine-tuning on ~3.4k high-quality trajectories from InfoSeek before on-policy RL (GRPO). Empirically, InfoFlow (3B/7B) outperforms strong open agents, even larger 32B ones, and is competitive with proprietary models on BrowseComp-Plus and general QA; ablations show the biggest drop without dual-agent RFT, with further declines when removing sub-goal shaping and hints, underscoring each component’s contribution.

**Strengths:**

- The paper clearly identifies and tackles the core bottleneck of deep, search-integrated reasoning—low reward density—and designs the system around increasing accuracy with less context, laying a principled target for optimization.

- The dual-agent architecture (Researcher + Refiner) is well-motivated and empirically effective: adding a 3B Refiner boosts success while cutting the Researcher’s context by ~45%, freeing capacity for higher-level planning.

**Weaknesses:**

1/ Despite gains, absolute accuracy on the hardest setting remains low (e.g., BrowseComp-Plus tops out around ~23% for 7B), suggesting substantial headroom and possible brittleness on deep-search tasks. Why do trajectories still fail most of the time, and what failure modes dominate (retrieval, planning, or grounding)?

2/ Evaluation relies on a single retriever (BM25) "for fair comparison." This controls a variable but also risks a retriever bottleneck driving the results. Would neural or hybrid retrieval change the ranking or reduce the need for hints/shaping? Please provide sensitivity to the retriever choice.

3/ The method’s strongest ablation effect is from the dual-agent RFT cold-start; without it, performance collapses (e.g., 23.2->10.2 on BrowseComp-Plus). This raises concern that the approach heavily depends on curated, high-quality trajectories; how robust is it when RFT data are noisy, scarce, or domain-shifted?

4/ Sub-goal shaping and off-policy hints densify rewards but may risk reward hacking or overfitting to annotation style. How are sub-goals validated, and can the policy succeed when hints are withheld at test time or when the sub-goal generator is imperfect?

**Questions:**

Please refer to the questions in Weaknesses.

---

> ### Author Response · Authors · 2025-11-25
> **Response to Reviewer GJJo (1)**
>
> > W1: Despite gains, absolute accuracy on the hardest setting remains low (e.g., BrowseComp-Plus tops out around ~23% for 7B), suggesting substantial headroom and possible brittleness on deep-search tasks. Why do trajectories still fail most of the time, and what failure modes dominate (retrieval, planning, or grounding)?
> * Thank you for highlighting this point. Following your suggestion, we performed a detailed error analysis on sampled failed trajectories from **BrowseComp-Plus** and **InfoSeek-Eval**.
> * **(1) Early decomposition and planning errors (18.1%).** A common source of failure is incorrect initial decomposition: the agent commits to a suboptimal search direction in the first few turns and fails to identify a viable entry point into the reasoning graph. Once the early plan drifts, later steps provide little meaningful information, and the trajectory never recovers.
> * **(2) Incomplete long-horizon reasoning (33.8%).** We observe cases where the agent forms a partially correct intermediate hypothesis but answers prematurely rather than fully validating all constraints. This reflects the inherent difficulty of sustaining multi-step search and verification over long horizons.
> * **(3) Retrieval errors (48.2%).** Nearly half of the failures stem from missing or low-quality evidence: the issued sub-queries fail to surface the needed document, or relevant pages are ranked too low by the retriever. Even when the reasoning direction is correct, repeated query rewrites often fail to recover the target information.
> * These findings clarify why accuracy remains far from saturated: the dominant bottleneck is not policy brittleness but rather the compound difficulty of long-horizon planning and high-recall retrieval.
>
> > W2: Evaluation relies on a single retriever (BM25) "for fair comparison." This controls a variable but also risks a retriever bottleneck driving the results. Would neural or hybrid retrieval change the ranking or reduce the need for hints/shaping? Please provide sensitivity to the retriever choice.
> * Thank you for raising this concern. Our original experiments used BM25 to keep the comparison across prior work controlled. We agree that only using BM25 can be a limiting factor, so we additionally evaluated InfoFlow with two modern dense retrievers: **E5** and **BGE-M3**.
>
> | Retriever | InfoFlow-3B | InfoFlow-7B |
> | --------- | ----------- | ----------- |
> | BM25      | 18.5        | 23.2        |
> | E5        | 21.3        | 25.7        |
> | BGE-M3    | 22.6        | 27.3        |
>
> These results show two points clearly:
> * **Retriever quality improves overall performance**, consistent with our error analysis where retrieval errors constitute a dominant failure source. Stronger retrievers substantially reduce this bottleneck.
> * **Retriever choice does not change the method ranking or the necessity of sub-goal shaping / hints.** InfoFlow benefits across all retrievers, and the relative trend remains consistent. Hints and shaping address long-horizon reasoning stability, not the retriever, and continue to provide value even when retrieval becomes stronger.

---

> ### Author Response · Authors · 2025-11-25
> **Response to Reviewer GJJo (2)**
>
> > W3: W3: The method’s strongest ablation effect is from the dual-agent RFT cold-start; without it, performance collapses (e.g., 23.2->10.2 on BrowseComp-Plus). This raises concern that the approach heavily depends on curated, high-quality trajectories; how robust is it when RFT data are noisy, scarce, or domain-shifted?
> We appreciate the reviewer’s careful reading and the concern regarding the sensitivity of our method to dual-agent RFT initialization. Below we clarify why (1) dual-agent RFT is not brittle to data scarcity or domain shift, and (2) remains effective even when the RFT data is intentionally made substantially noisier.
> * **Dual-agent RFT is a stable and general initialization strategy, not dependent on scarce or handcrafted data.** The RFT trajectories in our framework do not rely on manually curated external supervision. They can be obtained either by sampling from the model itself or from stronger open-source models. In practice, we found both sources sufficient to yield successful trajectories; therefore, the concern of “scarce or domain-shifted” RFT data does not arise. Moreover, jointly training the researcher and refiner agents stabilizes the initialization: the refiner is aligned to the researcher’s distribution, reducing drift during later RL and improving data efficiency compared to single-agent RFT.
> * **Robustness to noise: dual-agent RFT cold-start remains effective even when RFT data is significantly noisier.** To test robustness, we removed our filtering step and used *all* 16K rollouts—including trajectories containing search shortcuts, partially incorrect decompositions, and other artifacts—rather than the 3.4K high-quality filtered subset. This represents a substantial increase in noise.
>
> | Model             | NQ   | TQ   | PopQA | HotpotQA | 2Wiki | Musique | Bamboogle | Avg  |
> | ----------------- | ---- | ---- | ----- | -------- | ----- | ------- | --------- | ---- |
> | **RFT3B-Noisy**   | 39.0 | 55.6 | 45.4  | 39.8     | 42.3  | 16.7    | 39.4      | 39.8 |
> | **RFT3B-Cleaned** | 40.4 | 57.6 | 45.0  | 41.6     | 43.5  | 18.0    | 37.9      | 40.6 |
>
> Despite the substantial noise, the dual-agent RFT initialization remains effective and stable across all benchmarks, with only a modest performance drop relative to the filtered set. This confirms that the dual-agent structure is not brittle: it can extract meaningful patterns even from noisy trajectories and still provides a reliable cold-start for subsequent RL.
>
> > W4: Sub-goal shaping and off-policy hints densify rewards but may risk reward hacking or overfitting to annotation style. How are sub-goals validated, and can the policy succeed when hints are withheld at test time or when the sub-goal generator is imperfect?
> Thank you for raising this important point. We address the concerns regarding (i) reward hacking, (ii) dependence on hints, and (iii) robustness to imperfect sub-goal annotations.
> * **Sub-goal shaping does not induce reward hacking.** In the **revised paper (Sec. 4.3.3)**, we include RL training dynamics comparing methods *with* and *without* sub-goal shaping. We report both the original final reward and the combined reward. The key observation is that both curves improve jointly rather than diverging. If reward hacking occurred, the shaped reward would increase while the final reward stagnates or degrades. Instead, we see consistent gains in both metrics, indicating that the agent is genuinely learning better search policies rather than exploiting annotation artifacts.
> * **Off-policy hints do not create inference-time dependence.** Hints are injected only as *guiding search queries* when the rollout stalls. They are not additional supervision tokens or reasoning instructions. As such, the agent simply observes successful search patterns during training. At test time, where no hints are provided, the model operates normally, and our BrowseComp-Plus / InfoSeek-Eval results demonstrate that performance does not collapse.
> * **Further analysis and clarification.** Deep-search tasks naturally follow a hierarchical structure: progressing horizontally requires satisfying multiple constraints per layer, and progressing vertically resembles multi-hop reasoning. While this structure provides a natural basis for sub-goals, naively treating *every* node as a reward target risks noise and misalignment. To avoid this, we deliberately select only the *most critical* nodes through a strong external model, rather than using all intermediate nodes. This acts as a validation filter that suppresses spurious or low-value sub-goals, reducing susceptibility to reward hacking. Importantly, this process is automated to avoid prohibitive manual cost.

---

> ### Author Response · Authors · 2025-11-28
>
> Dear Reviewer GJJo,
>
> Thank you once again for your valuable comments on our submission. As the discussion phase is approaching its end, we would like to kindly confirm whether we have sufficiently addressed all of your concerns (or at least part of them). Should there be any remaining questions or areas requiring further clarification, please do not hesitate to let us know.
>
> We sincerely look forward to your feedback.

---

### Author Response · Authors · 2025-11-30
**Global Response to All Reviewers and AC**

We thank all reviewers for their time and constructive feedback. We are encouraged by the recognition of our work’s **effectiveness on complex information-seeking tasks (R1, R3), the novelty of the dual-agent framework (R1, R2), and the comprehensive analysis provided (R3, R4).**

During the rebuttal period, we have conducted extensive additional experiments to address concerns regarding **generalization, resource efficiency, and method robustness.** Below is a summary of our major revisions and response efforts.

### 1. Summary of New Experiments & Revisions
To address concerns about the scope, cost, and robustness of our method, we have added the following experiments:

*   **Generalization to Training Datasets without Native Trees and Costly APIs (R2, R3, R4):** To address concerns that our method is coupled to InfoSeek training datasets' specific annotations, we applied our pipeline to **HotpotQA** (which lacks explicit tree structure) using a standard open-source model (**Qwen3-8B**) to induce sub-goals and hints for RL training.
    *   **Result:** Our method achieved a **+1.6 average improvement** across 7 QA tasks compared to the baseline (w.o. hints and sub-goals). This confirms that our approach is **dataset-agnostic** and **does not rely on proprietary, high-end APIs** to yield gains.

*   **Robustness to Noisy Data (R1):** We stressed-tested the Dual-Agent RFT initialization by training on unfiltered, noisy trajectories (increasing data size from 3.4k to 16k).
    *   **Result:** The method maintained strong performance (Avg 39.8 vs 40.6), proving that the dual-agent structure is robust and **not brittle** to data noise or scarcity.

*   **Robustness to Retriever Choice (R1):** We evaluated InfoFlow with modern dense retrievers (E5, BGE-M3) in addition to BM25.
    *   **Result:** While stronger retrievers improve absolute numbers (e.g., +4.1% on InfoFlow-7B), the relative performance ranking remains consistent, and our error analysis confirms that retrieval quality is a primary bottleneck (48.2% of failures), which our method helps mitigate but benefits from improving.

*   **Efficiency & Cost Analysis (R3, R4):** We provided a quantitative evaluation of inference cost comparison under identical compute (8xH100).
    *   **Result:** We clarified that the **Dual-Agent framework is actually faster (10.2 min)** than the Single-Agent baseline (12.2 min). This is because the dual-agent design enables context compression, avoiding the quadratic attention costs associated with the single-agent’s accumulating context.

*   **Mechanism Validation (Hints & Rewards) (R1, R3):**
    *   **Hint Effectiveness:** We demonstrated that hint injection improves success rates even before RL (33.7 $\to$ 37.9), validating that they guide exploration effectively.
    *   **No Reward Hacking:** We added **RL training dynamics in sec. 4.3.3** showing that shaped rewards and final task rewards improve jointly, confirming the agent is learning the task, not exploiting the reward signal.

### 2. Restating Core Contributions
We believe our work addresses a critical bottleneck in RL-based deep search:

*   **Addressing Reward Sparsity:** We identify that reward density is the fundamental hurdle in long-horizon deep search. We propose a generalizable framework to densify rewards via sub-goal shaping and failure-guided hints.
*   **Effective Dual-Agent Architecture:** We demonstrate that separating "Planning/Reasoning" from "Synthesis/Refinement" not only stabilizes optimization but also improves inference efficiency by preventing context bloat.
*   **Open & Reproducible:** We have updated the anonymous repository with full code and data processing scripts. We are committed to open-sourcing the **Dual-Agent RFT data, Training Code, and Evaluation Pipeline** to facilitate future research in agentic search.

We have incorporated these new results and analyses into the revised manuscript. We hope these efforts resolve the remaining concerns, and we look forward to further discussion.

---

### Note · Authors · 2026-01-04

I have read and agree with the venue's withdrawal policy on behalf of myself and my co-authors.